# Chlorophyll Inhibits the Digestion of Soybean Oil in Simulated Human Gastrointestinal System

**DOI:** 10.3390/nu14091749

**Published:** 2022-04-22

**Authors:** Xiao Wang, Yuanyuan Li, Suxia Shen, Zhaotian Yang, Haifeng Zhang, Yan Zhang

**Affiliations:** 1College of Food Science and Nutritional Engineering, China Agricultural University, Beijing 100083, China; echowang2014@163.com (X.W.); liyuanyuan.1992@163.com (Y.L.); shen_sx573776028@163.com (S.S.); yang18515379662@163.com (Z.Y.); 2National Engineering Research Center for Fruits and Vegetables Processing, Ministry of Science and Technology, Beijing 100083, China; 3Key Laboratory of Fruits and Vegetables Processing, Ministry of Agriculture and Rural Affairs, Beijing 100083, China; 4Academy of Agricultural Planning and Engineering, Key Laboratory of Agro-Products Postharvest Handing, Ministry of Agriculture and Rural Affairs, Beijing 100125, China; 5Precision Nutrition (Shenzhen) Technology Co., Ltd., Shenzhen 518083, China; zhanghaifeng@genomics.cn

**Keywords:** chlorophyll, pheophytin, gastrointestinal digestion, pancreatic lipase

## Abstract

Nowadays, much available processed and highly palatable food such as cream products and fried and convenient food, which usually showed a high energy density, had caused an increase in the intake of dietary lipids, further leading to significant growth in the prevalence of obesity. Chlorophyll, widespread in fruits and vegetables, was proven to have beneficial effects on alleviating obesity. This study investigated the effects of chlorophyll on the digestive characteristics of lipids under in vitro simulated adult and infant gastrointestinal systems. Chlorophyll decreased the release rate of free fatty acid (FFA) during in vitro adult and infant intestinal digestion by 69.2% and 60.0%, respectively. Meanwhile, after gastrointestinal digestion, chlorophyll changed the FFA composition of soybean oil emulsion and increased the particle size of oil droplets. Interestingly, with the addition of chlorophyll, the activity of pancreatic lipase was inhibited during digestion, which may be related to pheophytin (a derivative of chlorophyll after gastric digestion). Therefore, the results obtained from isothermal titration calorimetry and molecular docking further elucidated that pheophytin could bind to pancreatic lipase with a strong affinity of (4.38 ± 0.76) × 10^7^ M^−1^ (*K_a_*), while the binding site was amino acid residue Trp253. The investigation not only explained why chlorophyll inhibited digestive enzyme activity to reduce lipids digestion but also provided exciting opportunities for developing novel chlorophyll-based healthy products for dietary application in preventing obesity.

## 1. Introduction

Dietary lipids collect the most highly concentrated energy in the human diet, which plays a significant role in helping achieve energy balance and controlling blood lipid levels by providing fatty acids [1,2]. At present, globalization has generated a sociocultural shift that has altered dietary habits [3], including increases in the consumption of animal offal, cream products, and fried and convenience foods, which are energy-dense [4,5]. However, excessive intake of lipids generates an imbalance between energy intake and consumption, leading to a significant increase in the prevalence of obesity, accompanied by a concomitant rise in the incidence of metabolic disorders, such as type 2 diabetes and non-alcoholic fatty liver disease [6]. Therefore, obesity has reached pandemic proportions worldwide, which greatly reduces the quality of life and lifespan on a global scale [7].

Hitherto, considerable research efforts have been keen on seeking interventions on diminishing the prevalence of obesity and related complications induced by excessive lipids intake [8]. Current strategies for interventions include reducing the amount of saturated fat, cholesterol, or trans-fatty acids in food to decrease unfriendly lipids intake [9,10,11,12,13] or regulating the digestion and absorption of lipids [14,15]. Fortunately, some researchers found that the intake of bioactive compounds from plant-based food, such as polyphenols and anthocyanins, acted synergistically and healthily to ameliorate obesity [16,17]. The possible mechanisms by which natural substances act against obesity include inhibiting the digestion and absorption of lipids [18], regulating the appetite and satiety [19], improving expenditure of energy [20], inhibiting lipids biogenesis [21], decreasing differentiation of pre-adipocytes [22], and increasing the breakdown of triglycerides (TAG) [23].

For instance, confirmatory evidence has reported that oil-in-water emulsion could form a physical barrier to block the contact between oil and lipase, thereby reducing the digestion of lipids [24]. According to Zhou, et al. [24], the combination of gliadin and proanthocyanidins was beneficial to the interfacial adsorption of emulsions, which prevented lipids digestion in vitro gastrointestinal models. Furthermore, studies revealed that curcumin, sinensetin, and *Sargassum horneri* inhibited the activity of pancreatic lipase to further prevent lipids absorption [25,26]. In addition, researchers also addressed that soybean-based liposomes could decrease the degree of lipids hydrolysis during in vitro digestion [27]. These results highlight the importance of natural substances in reducing lipids digestion. 

As one of the most abundant liposoluble pigments in nature, chlorophyll is broadly distributed in fruits, vegetables, and plant-derived foods [28]. Recently, it was established that chlorophyll and its derivatives could prevent diabetes and certain types of cancer [29]. In our previous research, intake of chlorophyll early in life effectively alleviated body weight gain, improved glucose tolerance, and decreased the level of plasma triglyceride (TC) and total cholesterol (TG), showing a beneficial effect on obese mice [30,31]. Nevertheless, it remains elusive whether chlorophyll and its derivatives could affect the digestion of lipids in the gastrointestinal tract. 

In the current work, we implemented in vitro adult and infant gastrointestinal digestion models to investigate the effects of chlorophyll on lipids digestion in-depth. Our work investigated the effects of chlorophyll on lipids digestion by determining the release of free fatty acids, the mean particle size of oil droplets, and the activity of lipase. The investigation not only provides new insights to explain the bioactivity of chlorophyll in lipids digestion but also opens exciting opportunities for developing novel chlorophyll-based healthy products for dietary application in preventing obesity.

## 2. Materials and Methods

### 2.1. Samples and Materials 

Chlorophyll extracts were prepared as previously reported by Li, et al. [32], and the concentration was 9 mg/mL. Soybean oil was obtained from Yihai Kerry Arawana Holdings Co., Ltd. (Beijing, China). Whey protein isolate (WPI) was purchased from Davisco Foods International Inc. (Le Sueur, MN, USA). Porcine pepsin (Sigma, P6887), bovine lipase (Sigma, L2254), porcine pancreatin (Sigma, P7545), bovine bile (Sigma, B8631), lipase from pancreatin (Sigma, L31126), and Nile red (Sigma, 72485) were obtained from Sigma-Aldrich (St. Louis, MO, USA). The determination of enzyme activities referred to the method of Minekus, et al. [33]. All chemicals and reagents were at or above the standard analytical grade.

### 2.2. Preparation of Initial Lipid Digestion Emulsions

An emulsifier solution containing 1.0% WPI in 5 mM PBS (pH 7.0) (*w*/*w*) was prepared and then stored overnight in the refrigerator at 4 °C to ensure complete hydration. First, 48 mg and 100 mg freeze-dried chlorophyll extracts were pre-dissolved in 20 g soybean oil and fully mixed in the dark. Then, the mixtures of soybean oil and chlorophyll extracts were homogenized with 1.0% WPI at 12,000 rpm for 3 min by using a high-speed blender (T18D S25, IKA, Staufen im Breisgau, Germany). A mixture of soybean oil without chlorophyll extract was also homogenized with 1.0% WPI in the same manner. The coarse emulsions contained 10% lipid phase and 90% emulsifier solution (*w*/*w*) and different concentrations of chlorophyll extracts. In addition, the coarse emulsions were homogenized 3 times at 380 bar with a nanometer high-pressure homogenizer (JNBIOJN-10HC, Guangzhou Juneng Nano & Bio Technology Co., Ltd., Guangzhou, China). The final concentrations of chlorophyll were 0, 0.24%, and 0.50% (*w*/*w*), based on 20 g soybean oil.

### 2.3. Model of Gastrointestinal Digestion In Vitro

In vitro gastrointestinal digestion models for adults and infants were established. The diet for infant is usually semi-liquid and remains in the mouth for a short time. Thus, all emulsions were carried out with two-stage static in vitro digestion independently, following internationally recognized methods [33,34] with slight modifications. 

The simulated gastric fluid (SGF) and simulated intestinal fluid (SIF) were obtained with NaCl, KH_2_PO_4_, NaHCO_3_, MgCl_2_(H_2_O)_6_, and (NH_4_)_2_CO_3_, according to the methods of Li, et al. [32]. To simulate gastric digestion, 7.5 mL of SGF, 1.6 mL of pepsin solution, and 5 μL of 0.3 M CaCl_2_ were mixed with 10 mL of each emulsion sample, and double distilled water was added to reach the final volume of 20 mL. Notably, the pH of SGF for adult and infant models was adjusted by 6 M HCl to 3.0 and 5.3, respectively. The final concentrations of pepsin in adult and infant digestive systems were 2000 U/mL and 268 U/mL, respectively. The food matrix was incubated for 120 min at 37 °C in the water bath oscillator (180 r/min). 

During intestinal digestion, 11 mL of SIF, 5 mL of pancreatic enzyme solution, 2.5 mL bile extract, and 40 μL of 0.3 M CaCl_2_ were added to each sample tube, and a final volume of 40 mL was achieved by adding double distilled water. In particular, the pH of SIF for adults and infants was adjusted by 6 M NaOH to 7.0 and 6.6, respectively. The final enzyme solution contained 2000 U/mL and 90 U/mL of pancreatic lipase, respectively. Meanwhile, the final bile salt concentration in adult and infant systems was ~10 mM and 3.1 mM, respectively. Then, all samples were incubated for 120 min in the water bath oscillator (165 r/min) at 37 °C. 

Abbreviations of different groups were as follows: AL, in vitro adult digestion without chlorophyll; AL-CL, in vitro adult digestion with lower dose of chlorophyll (0.24%, *w*/*w*); AL-CH, in vitro adult digestion with higher dose of chlorophyll (0.50%, *w*/*w*); EL, in vitro infant digestion without chlorophyll; EL-CL, in vitro infant digestion with lower dose of chlorophyll (0.24%, *w*/*w*); EL-CH, in vitro infant digestion with higher dose of chlorophyll (0.50%, *w*/*w*).

### 2.4. Characterization of Particle Size and ζ-Potential

The droplet sizes of initial emulsions and emulsions after different digestion stages were determined by Mastersizer 3000 (Malvern Instrument Ltd., Malvern, UK). The refractive index of soybean oil and dispersed phase were 1.476 and 1.333, respectively. The digested sample of 0.2 mL was diluted to an appropriate multiple to eliminate the multiple scattering effect. The particle size of emulsion droplets was reported as surface weighted mean diameter (*d*_3,2_) and was calculated by $d_{3,2}=\sum n_{i}d_{i}{}^{3}/\sum n_{i}d_{i}{}^{2}$, where *n_i_* is the number of particles, and *d_i_* is the diameter of emulsion droplets.

Zetasizer ZEN 3700 (Malvern Instrument Ltd., Malvern, UK) was used to measure the ζ-potential. All samples were diluted with Milli-Q water, SGF, or SIF to obtain droplet concentration of about 0.005% (*w*/*w*). The ζ-potential measurement was calculated based on at least five readings for an individual sample.

### 2.5. Observation of Microstructure 

A confocal laser scanning microscope (Zeiss, Jena, Germany) was used to observe the microstructures of initial lipids digestion emulsions and emulsions after different digestion phases. Firstly, 200 μL of each sample was fully mixed with 10 μL of Nile red solution (10 mg/mL in ethanol). Then, an appropriate aliquot of sample was placed on a microscope slide, covered by a glass coverslip, and sealed off from air. All procedures were carried out in the dark. Acquisition and processing of all images were conducted by Zen 3.4 software (Zeiss, Thornwood, NY, USA).

### 2.6. Analysis of Free Fatty Acid (FFA) Release and Hydrolysis Kinetics 

A pH-stat automatic titration system (904 Titrando, Herisau, Switzerland) was used to monitor the release of FFA at 0, 5, 10, 15, 30, 45, 60, 90, and 120 min during intestinal digestion. Notably, 0.1 M NaOH was added to continuously maintain the pH of intestinal digestion system at 7.0 (adult) and 6.6 (infant). The consumed volume of 0.1 M NaOH solution was recorded, and the release percentage of FFA in the system was calculated by Equation (1) [35]: (1)$$FFA\%=100\times \frac{V_{NaOH}\times m_{NaOH}\times M_{Lipid}}{2\times W_{Lipid}}$$
where *V_NaOH_* (L) is the volume of NaOH consumed during intestinal digestion, *m_NaOH_* (M) is the molarity of NaOH, *M_Lipid_* (g/mol) is the average molecular weight of soybean oil, and *W_Lipid_* (g) is the total weight of initial lipids in the digestion system. Generally, it was assumed that one molecule of triglyceride produced two molecules of fatty acids.

The release of FFA gradually increased with the extension of intestinal digestion time, finally achieving the total FFA release (*Φ_max_*). The kinetic parameters for FFA release were calculated by using Eqaution (2) [36]:(2)$$\ln\left[(\Phi_{max}-\Phi_{t})/\Phi_{max}\right]=-kt+b$$
where *k* is the first-order rate constant for FFA release (s^−1^), while *t* is the time point (s) of digestion.

### 2.7. Determination of Fatty Acid Composition

A certain volume of sample was accurately taken in the hydrolysis tube with 4 mL of acetyl chloride/methanol (1:10, *v*/*v*), 1 mL of 1,2,3-Triundecanoyl Glycerol (C11:0) internal standard solution (1.0 mg/mL), and 1 mL of hexane. The test tube with a cap was then put in a water bath at 80 °C for 2 h. After cooling, 5 mL of 7% K_2_CO_3_ solution was added to the tube and shaken evenly. The sample was centrifuged at 1000× *g* r/min for 5 min and filtered by a 0.2 μm filter membrane.

Fatty acids were identified and qualified by 7890 series gas chromatography (Agilent Co., Santa Clara, CA, USA), equipped with a flame ionization detector and a capillary gas chromatography column (DB-23 60.0 m × 250 μm × 0.25 μm). He was supplied as the carrier gas at a flow rate of 2.0 mL/min, and the split ratio was 300. The initial temperature of the oven was set at 40 °C, maintained for 1 min, and then increased to 220 °C with a rate of 5 °C/min for 2 min. Afterward, the temperature was increased to 260 °C and finally held at 260 °C for 2 min. The temperatures of the injector and detector were set to 260 °C and 270 °C, respectively. In addition, the injection volume was 1 μL. Finally, fatty acids were quantified by comparing the relative retention time and the total correction peak area of fatty acid methyl standard.

### 2.8. Determination of Pancreatic Lipase Activity

During intestinal digestion, the activity of pancreatic lipase was determined at 0, 20, 60, and 120 min. The activity of pancreatic lipase during oil digestion was measured by lipase assay kit (Nanjing Jiancheng Bioengineering Institute, Nanjing, China).

### 2.9. Molecular Docking

The interaction between chlorophyll-derivative pheophytin and pancreatic lipase was further supported by molecular docking analysis. The structure of pancreatic lipase was obtained from the RCSB PDB website (PDB ID: 1ETH), and the structure of pheophytin was modified and optimized by Discovery Studio (Version 4.5), according to Li, et al. [37]. Then, the interactions between pheophytin and pancreatic lipase were processed by AutoDock software (Version 4.2.6). AutoDock provided the position, orientation, and conformation of the ligand in the active site in detail. The docking attitude was generated by the combination of an evolutionary algorithm and simulated annealing search. When each simulated annealing search began, the input ligand conformation (including its position, orientation, and torsions) could be randomly changed. Therefore, each search could start with a different ligand conformation. PyMOL Molecular Graphics System (Version 2.0.6, Schrödinger, New York, NY, USA, LLC) was used to analyze the results of molecular docking.

### 2.10. Fluorescence Titration

The fluorescence was recorded by a Cary Eclipse Spectrophotometer (Agilent Co., Santa Clara, CA, USA). About 1 mL of pancreatic lipase solution (10 μM) was added to the cell. Then, 2 μL of pheophytin solution (0.23 mM) was titrated into one cell at a time. Notably, pancreatic lipase and pheophytin were both dissolved into 20 mM Tris-HCl (pH 7.0). The fluorescence excitation wavelength was set to 280 nm, and the emission spectrum was recorded at 290 to 500 nm. All fluorescence experiments were conducted in triplicate. 

To evaluate the binding stoichiometry induced by pheophytin, the Stern–Volmer equation (Equation (3)) [38] was used as follows:(3)$$\log\left[\left(F_{0}-F\right)/F\right]=\log K_{a}+n\log\left[Q\right]$$
where *F*_0_ and *F* are the fluorescence intensities in the absence and presence of quencher (pheophytin), respectively, *n* is the number of binding sites, and [*Q*] is the concentration of quencher (pheophytin). The value of *K_a_* can be obtained from the intercept of the plot of log [(*F*_0_ − *F*)/*F*] versus log [*Q*], whereas the binding site *n* was the slope. 

### 2.11. Circular Dichroism (CD) Spectra 

CD spectroscopy for pancreatic lipase in the absence and presence of pheophytin was recorded on a PiStar-180 spectrometer (Applied Photophysics Ltd., Leatherhead, UK) equipped with a path length of 1 mm quartz cell. For recording far-UV region (200–260 nm), CD spectra were scanned with five replicates under a scanning speed of 50 nm min^−1^ at 310 K and the bandwidth was set as 1 nm. Pancreatic lipase and its complex (molar ratio of lipase to pheophytin was 1:1) were diluted in buffer solution (20 mM Tris-HCl, pH 7.0). The buffer was treated as blank background. Afterward, secondary structures fractions of lipase and its complex were calculated by CNDD software.

### 2.12. Isothermal Titration Calorimetry (ITC)

All ITC experiments were performed using a Nano-ITC II instrument (TA Instruments, New Castle, DE, USA) at 310 K. The concentration of pheophytin and pancreatic lipase was 0.3 mM and 40 μM, respectively, in 20 mM Tris-HCl (pH 7.0). Prior to titrations, all solutions were filtered by a 0.45 μm filter membrane and then thoroughly degassed under 550 mmHg for 30 min. Next, 200 μL of buffer or pancreatic lipase solution was put into the sample cell. Furthermore, it was necessary to ensure that there were no bubbles in the syringe after loading pheophytin. Once the baseline was stable, 20 aliquots of pheophytin solution (2.5 μL each) could be titrated to the lipase solution.

### 2.13. Statistical Analysis

All experiments were repeated at least three times. All data were presented as mean ± standard deviation (SD). The results obtained were expressed with SPSS 22.0 software (SPSS Statistics, New York, NY, USA) using Duncan’s multiple range test, with a significance level of *p* < 0.05. All graphs were drawn using Origin 9.0 software (Origin Lab, Northampton, MA, USA).

## 3. Results and Discussion

### 3.1. Effects of Chlorophyll on Soybean Oil Digestion

#### 3.1.1. Changes in FFA Release and Fatty Acids Composition

The hydrolysis degree and rate of lipids can be determined by recording the release rate of FFA during in vitro intestinal digestion. Upon entry into the small intestine, dietary triglycerides are emulsified by bile acids and then broken down by pancreatic lipase to produce 2-monoacyglycerol and fatty acids [39]. As shown in Figure 1, interestingly, the release of FFA during intestinal digestion decreased with the presence of chlorophyll both in the adult and infant digestion systems. In the first 30 min, both AL-CL and AL-CH had a rapid production of FFA to a maximum level, then followed by no obvious increase (Figure 1A). In contrast, the FFA released from AL still increased following digestion time points. The maximum FFA released from AL-CL (14.8 ± 2.2%) and AL-CH (10.9 ± 1.3%) decreased by 58.1% and 69.2%, respectively, compared with AL (35.4 ± 4.5%) (Figure 1A). Likewise, the final release of FFA in EL-CL (4.8 ± 1.5%) and EL-CH (4.2 ± 2.0%) decreased by 54.3% and 60.0%, respectively, versus EL (10.5 ± 2.3%) (Figure 1B). These results indicated that in the presence of chlorophyll, maybe only a small amount of lipase molecules moved from the aqueous phase and attached to the surfaces of oil droplets, further hydrolyzing triacylglycerol molecules [15]. Recent studies found that when adding 0.75% *w*/*w* nanocellulose to a high-fat food, the hydrolysis rate of FFA by triglycerides was reduced by 48.4% in simulated gastrointestinal digestion [40]. Here, the present work presented for the first time that chlorophyll exhibited a greater impact on the digestion of lipids in vitro, and it could even be a better choice to reduce the release rate of FFA.

To further investigate the role of chlorophyll in lipids digestibility, Equation (2) was used to analyze the linear relationships between FFA release and digestion time. Table 1 provided the curve-fitting parameters, including apparent rate constants (*k*) and regression coefficients. All FFA release curves had good linear correlations (all above 0.9000). The presence of chlorophyll made the apparent rate constant smaller, indicating that chlorophyll could reduce the degree of hydrolysis and the rate of lipids digestion. A similar result reported that cellulose-nanoparticles-stabilized emulsions with a higher FFA release rate (0.029 min^−1^) showed a great impact on lipolysis rate [41].

Pancreatic lipase mainly hydrolyzes fatty acids at the sn-1,3 position of triglycerides in the small intestine [39]. Does the presence of chlorophyll affect pancreatic lipase-mediated lipids digestion, thereby modulating the composition of fatty acids? The results shown in Table 2 identified that the most abundant fatty acid was C18:2, followed by C18:1 and C16:0 in all groups. The relative amount of C18:2 and C18:1 released in AL-CL and AL-CH during digestion was much lower than that of AL (*p* < 0.05), while there was a similar tendency observed under infant digestion. Compared with AL, the relative amount of C16:0 and C18:0 in AL-CH significantly increased (*p* < 0.05), suggesting that in the presence of chlorophyll, C16:0 and C18:0 might prefer to transfer to micelles to a higher degree in adult digestion systems. Nevertheless, no significant difference was observed in the relative content of C16:0 and C18:0 among EL, EL-CL, and EL-CH (*p* > 0.05). Moreover, following the relative content of C18:3 (α-linolenic acid, ALA), a significant increase was observed in both adult and infant digestion systems (*p* < 0.05) in the presence of chlorophyll. A recent study showed that ALA could improve blood lipid profiles by decreasing levels of TG, TC, and LDL, which was beneficial to obesity, hyperlipidemia, and cardiovascular disease [42]. Therefore, these results might also provide implications for chlorophyll to further affect the luminal absorption of fatty acids by intestinal epithelial cells.

#### 3.1.2. Evolutions of Oil Droplet Size and Surface Charge

After entering the small intestine, dietary lipids are further emulsified by bile acids to form mixed micelles with smaller droplet sizes, which greatly enhance lipolysis by pancreatic lipase at the oil–water interface [43]. In addition, the particle size of oil droplets can further affect the intestinal uptake of lipids by the enterocytes, so we measured the particle size of oil droplets in initial emulsions and emulsions after different adult and infant digestion stages (Figure 2A,B). Meanwhile, the microstructures of all samples were obtained using confocal laser microscopy (Figure 2C). The observed difference in the mean particle size of oil droplets among initial emulsions was not significant (*p* > 0.05), despite containing different concentrations of chlorophyll. 

During gastric digestion, the mean particle sizes of oil droplets in adult and infant digestion systems significantly increased versus initial emulsions (*p* < 0.05) (Figure 2A,B). In addition, as depicted in Figure 2C, each mixed micelle still showed apparent aggregation or bridging state, and large oil droplets appeared, especially in the addition of chlorophyll groups. These results indicated that the flocculation or aggregation of oil droplets might be due to the hydrolysis of proteins adsorbed in all emulsions under low pH gastric conditions [44].

It could be seen that the mean particle sizes of oil droplets in AL-CL and AL-CH were markedly larger than those of AL (*p* < 0.05), Figure 2A. Consistently, a similar trend was seen in the in vitro infant digestion systems (Figure 2B). In particular, the oil droplet sizes of all samples in adult systems (2.59 to 17.26 μm) were much higher than those in infant systems (0.45 to 0.73 μm). This may be owed to the different activities of pepsin in the stomachs of adult and infant systems, which led to different degrees of protein hydrolysis [32].

Upon intestinal digestion, as shown in Figure 2A, with the continuous hydrolysis of lipids, the mean particle sizes of oil droplets in adult digestion systems significantly decreased versus in gastric digestion (*p* < 0.05). Nevertheless, in infant digestion systems (Figure 2B), the mean particle sizes markedly increased versus in gastric digestion (*p* < 0.05). Additionally, Figure 2C also presented that large oil particles in adult systems disappeared after intestinal digestion, and the bridging effect among particles weakened, whereas oil droplets in infant systems still showed a large aggregation state. The reason for this discrepancy might be that the concentration of pancreatic lipase in infant intestinal digestion was lower than that of adult systems, and the optimal conditions of enzyme reaction had not been reached [34].

The effects of chlorophyll on the particle size of oil droplets during intestinal digestion were further compared. Interestingly, the particle size of oil droplets in AL-CH (0.88 ± 0.06 μm) was relatively larger than that in AL (0.45 ± 0.03 μm) (*p* < 0.05). Similarly, it was obvious that the mean particle size of oil droplets in EL-CH was significantly larger than in EL (*p* < 0.05). This could be attributed to the liposolubility of chlorophyll enhancing the aggregation of chlorophyll with lipid droplets or the interactions between chlorophyll and lipase, which weakened the lipolysis during digestion. Therefore, chlorophyll could increase the mean particle size of oil droplets after in vitro gastrointestinal digestion.

Changes in the ζ-potential of all emulsions during different gastrointestinal digestion stages were exhibited in Figure 3. The ζ-potential values of initial emulsions varied between −17.2 mV and −21.5 mV, and the ζ-potential value significantly decreased in the presence of chlorophyll (*p* < 0.05), which suggested that electronegative chlorophyll had a slight to moderate negative charge [37]. In adult gastric digestion (Figure 3A), due to the electrostatic shielding effect under conditions of strong acidity and high concentration of ions, the stability of emulsions decreased versus the initial stage. In addition, pepsin might hydrolyze the interface protein film of emulsion droplets, which weakened the electrostatic repulsion to generate lipids aggregation. The ζ-potential of AL-CL and AL-CH was significantly lower than AL (*p* < 0.05). Still, compared to gastric stage with lower negative charge, there was a dramatic increase in negative charge after intestinal stage (*p* < 0.05) in adult digestion systems. Conversely, in infant digestion systems (Figure 3B), all emulsions still presented strong negative charge during gastric digestion, indicating that poor hydrolysis ability of pepsin in the environment of infant digestion systems [34]. Furthermore, during intestinal digestion, the lower concentration of pancreatic lipase and bile salt in infant digestion systems may account for the decrease in surface negative charge after intestinal digestion. Nevertheless, it was worth noting that chlorophyll could increase the absolute value of ζ-potential both in adult and infant digestion systems.

### 3.2. Mechanism of Chlorophyll Inhibiting Lipid Digestion

#### 3.2.1. Effect of Chlorophyll on the Activity of Pancreatic Lipase during Intestinal Digestion

To elucidate why chlorophyll inhibited soybean oil digestion, the activity of pancreatic lipase was measured during intestinal digestion, as depicted in Figure 4. In adult and infant simulated digestion systems, the initial lipase activity was about 2000 U/L and 800 U/L, respectively. At 120 min, the lipase activity of AL-CH (785.1 ± 73.3 U/L) was significantly lower than that in AL (1134.8 ± 72.6 U/L) (*p* < 0.05) (Figure 4A). It could be seen that the presence of chlorophyll decreased the activity of pancreatic lipase by 30.8% during lipids intestinal digestion. Consistently in infant intestinal digestion systems, the final lipase activity of EL-CH (732.1 ± 14.5 U/L) remained lower than EL (749.7 ± 5.5 U/L) (Figure 4B). As our previous study reported, chlorophyll preferred to lose its central Mg^2+^ mental to generate pheophytin under lower pH conditions during in vitro adult and infant gastric digestion [32]. Consequently, according to the existing results, we speculated that pheophytin may be a key substance affecting lipase activity during lipids digestion. The role of natural bioactive compounds as pancreatic lipase inhibitory in regulating intestinal lipolysis had been proposed by Huang, et al. [45]. Huang, et al. [45] revealed that grapefruit, pomelo, and kumquat peel extracts rich in flavonoids demonstrated the potential of inhibiting pancreatic lipase effectively, while IC_50_ ranged from 216.07 to 292.11 μg/mL. Kim, et al. [46] also found that *Phaseolus multiflorus var. albus* Bailey extract could inhibit the activity of pancreatic lipase with an IC_50_ value of 1.68 mg/mL. Moreover, we determined that the IC_50_ value of pheophytin was 81.1 μg/mL, suggesting pheophytin might have a more substantial inhibitory effect on pancreatic lipase, which could be attributed to the stronger binding ability of pheophytin to lipase.

These findings provided the first evidence that pheophytin could be a potent inhibitor of a key enzyme in lipids digestion in vitro. Hence, we designed experiments to demonstrate why pheophytin affected pancreatic lipase activity. 

#### 3.2.2. Effect of Pheophytin on Structure–Activity Relationships of Pancreatic Lipase

Firstly, fluorescence spectroscopy was conducted to obtain further information on whether pheophytin interacted with pancreatic lipase. The characteristic fluorescence of protein was mainly conditioned by tryptophan residues due to its sensitivity to the microenvironment surrounding the fluorescence residues [47]. As depicted in Figure 5A, under the excitation of 280 nm, pancreatic lipase showed a strong fluorescence emission band at 353 nm in the absence of pheophytin. With the increase in the pheophytin concentration, the fluorescence intensity of pancreatic lipase decreased gradually at about 353 nm, which could be attributed to the interaction between pheophytin and pancreatic lipase, such as the excited-state reaction, molecular rearrangement, energy transfer, ground-state complex formation, and the collision quenching process [48]. Meanwhile, in the presence of pheophytin, the wavelength of the maximum fluorescence intensity of lipase gradually blue-shifted from 353 nm to 350 nm, suggesting the hydrophobicity around the protein chromophore group might be strengthened and the tryptophan residues of protein were exposed from the hydrophilic environment to the hydrophobic environment [47]. 

As shown in Appendix A, the binding constant (*K_a_*) and the number of binding sites (*n*) between pheophytin and lipase calculated from Equation (3) were (0.2392 ± 0.0025) × 10^4^ M^−1^ and 1.39562 ± 0.0135, respectively, elucidating that one pancreatic lipase molecule could bind to only one pheophytin molecule. These results corroborated the ideas of Wu, et al. [18], who demonstrated that quercetin-3-rhamnoside, as a pancreatic lipase inhibitor, could bind to pancreatic lipase, and the binding molar ratio was 1:1. Therefore, the results of fluorescence titration revealed that pheophytin could indeed quench the fluorescence of pancreatic lipase and bind to pancreatic lipase.

Isothermal titration calorimetry (ITC), based on accurate measurement of the heat released or absorbed during the formation of the complex, is considered to be the most sensitive and reliable tool to obtain the thermodynamic binding parameters of proteins to small molecules [49]. To further characterize the binding of pheophytin to pancreatic lipase, ITC was then conducted to determine the binding constant (*K_a_*), reaction stoichiometry (*n*), and enthalpy (Δ*H*^0^). In this study, the concentrated pheophytin solution (0.3 mM) in the syringe was progressively diluted into the sample cell containing the pancreatic lipase solution (40 μM) at 37 °C. The positive peaks seen in Figure 5B showed that the binding of pheophytin to pancreatic lipase was an exothermic reaction, while the interaction was spontaneous. With the increase in the molar ratio of pheophytin to lipase, the heat for each injection decreased gradually. An appropriate binding model was selected to fit the ITC data, giving an apparent equilibrium constant that *K_a_* was (4.38 ± 0.08) × 10^7^ M^−1^, corresponding to strong interaction [50]. The fitting result of binding site was 1.00 ± 0.01, indicating that a pancreatic lipase molecule was bound to a pheophytin molecule to perform a Phe–Lipase complex. In addition, the negative value of Δ*H*^0^ (−1137.01 ± 0.76 kJ/mol) indicated that hydrophobic interaction could be predominant in the interaction between pheophytin and pancreatic lipase. Interestingly, Ghayour, et al. [51] also demonstrated that hydrophobic interaction was the major driving force in the binding of proteins to hydrophobic compounds, such as lysozyme and astaxanthin.

To capture more information about the interaction between pancreatic lipase and pheophytin, circular dichroism (CD) experiments were carried out. Figure 6A showed that there was almost no change in the CD spectrum of pancreatic lipase in the presence of pheophytin, indicating that the interaction between pancreatic lipase and pheophytin hardly changed the secondary structure of the protein. Similarly, a recent study showed that the interaction between pancreatic lipase and hesperidin could not change the secondary conformation of lipase [45]. Meanwhile, the contents of α-helix, β-sheet, β-turn, and random coil depicted in Figure 6B were simulated by CDNN software. Upon interaction, the protein conformational analysis suggested that the Phe–Lipase complex had a slight increase in the content of α-helix and a decrease in the content of β-sheet, β-turn, and random coil.

#### 3.2.3. Molecular Docking Simulation

The possible mechanism for the structure–activity relationships between receptors and ligands can be studied by molecular docking simulation [52]. 

As shown in Figure 7, the interaction between pheophytin and pancreatic lipase was investigated by molecular docking methods. The binding was dominated by hydrophobic interaction, and the binding site was amino acid residue Trp253, which belongs to hydrophobic pockets of pancreatic lipase. The predicted binding energy of pheophytin and pancreatic lipase was −2.88 ± 0.82 kcal mol^−1^. According to previous reports, amino acid residues Ser153, Phe78, Phe216, Ile79, Tyr115, His264, Lys271, Thr68, and Pro177 played crucial roles in the binding of pancreatic lipase to small bioactive molecules [53,54]. Hence, our results reported for the first time that the conformation of pheophytin binding to pancreatic lipase may be due to the strong hydrophobic interaction, while the binding site was amino acid residue Trp253. Combined with all the above results, the interaction between pheophytin and pancreatic lipase could inhibit lipase activity, which was beneficial to reducing lipids digestion of soybean oil under in vitro simulated gastrointestinal systems. 

## 4. Conclusions

The investigation presented the effects of chlorophyll and its derivatives on the kinetics of lipids digestion under in vitro stimulated adult and infant gastrointestinal conditions for the first time. The results showed that the addition of chlorophyll decreased the release rate of free fatty acid and changed the fatty acid composition of soybean oil emulsions under in vitro gastrointestinal digestion. In addition, it was found that pheophytin, a derivative of chlorophyll after gastric digestion, was bound to pancreatic lipase to inhibit pancreatic lipase activity during intestinal digestion. Moreover, the increase in the mean particle size of oil droplets induced by chlorophyll in the intestine stage may affect the latter absorption and transport of fatty acids in intestinal epithelial cells, which need to be further explored in vivo with animals or human models. The results provided mechanistic insights into the role of chlorophyll and its derivatives in reducing lipids digestion and introduced exciting opportunities for developing novel chlorophyll-based healthy products for preventing obesity.

## Figures and Tables

**Figure 1 nutrients-14-01749-f001:**
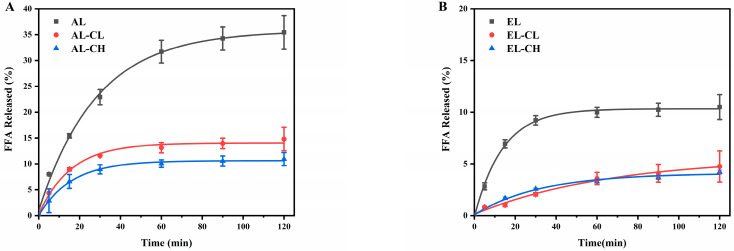
Free fatty acids (FFA) released from different emulsions during in vitro intestinal digestion of (**A**) adult and (**B**) infant. Data were shown as mean ± SD of three replicates. AL, in vitro adult digestion without chlorophyll; AL-CL, in vitro adult digestion with lower dose of chlorophyll (0.24%, *w*/*w*); AL-CH, in vitro adult digestion with higher dose of chlorophyll (0.50%, *w*/*w*); EL, in vitro infant digestion without chlorophyll; EL-CL, in vitro infant digestion with lower dose of chlorophyll (0.24%, *w*/*w*); EL-CH, in vitro infant digestion with higher dose of chlorophyll (0.50%, *w*/*w*).

**Figure 2 nutrients-14-01749-f002:**
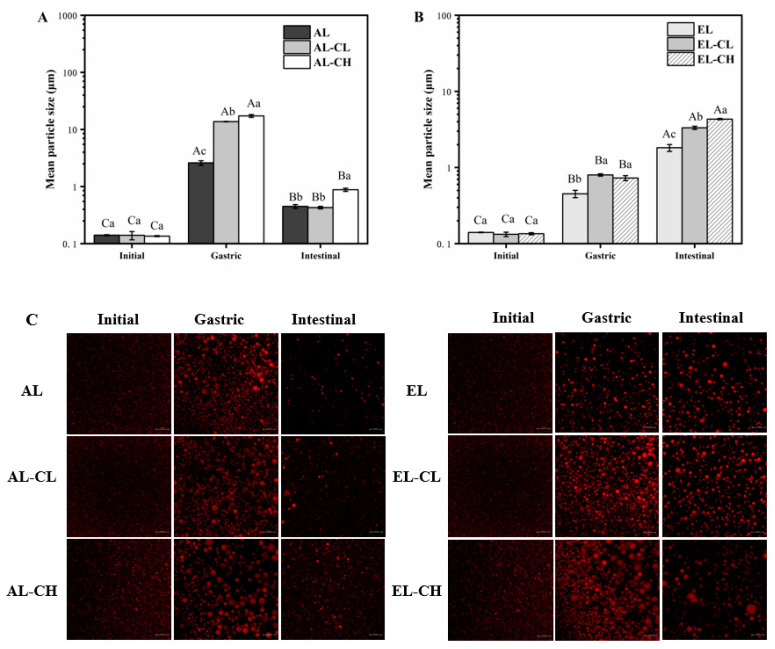
Mean particle size of initial emulsions and emulsions after different digestion stages of (**A**) adult and (**B**) infant and (**C**) microstructure images of emulsions after exposed to different digestion stages (scale bar of 10 μm in all images). Data were shown as mean ± SD of three replicates. (**A**–**C**) Different capital letters indicated significant differences (*p* < 0.05) in the same group (with same legend) at initial, gastric, and intestinal stages of digestion. a–c Different lowercase letters indicated significant differences (*p* < 0.05) among different groups (with different legends) at the same stage of digestion. The same letter indicated no significant difference (*p* > 0.05).

**Figure 3 nutrients-14-01749-f003:**
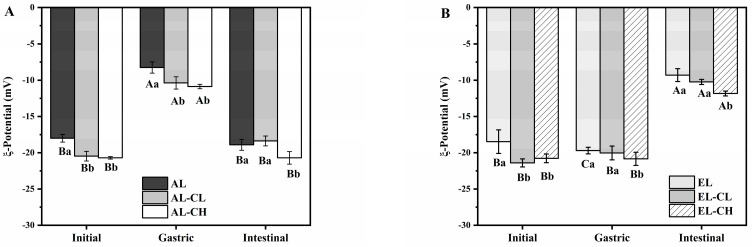
Zeta-potential of initial emulsions and emulsions after different digestion stages of (**A**) adult and (**B**) infant. Data were shown as mean ± SD of three replicates. A-C Different capital letters indicated significant differences (*p* < 0.05) in the same group (with same legend) at initial, gastric, and intestinal stages of digestion. a-c Different lowercase letters indicated significant differences (*p* < 0.05) among different groups (with different legends) at the same stage of digestion. The same letter indicated no significant difference (*p* > 0.05).

**Figure 4 nutrients-14-01749-f004:**
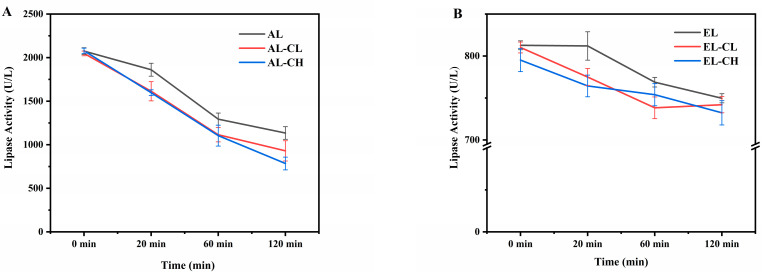
Lipase activity during in vitro (**A**) adult and (**B**) infant intestinal digestion. Data were shown as mean ± SD of three replicates.

**Figure 5 nutrients-14-01749-f005:**
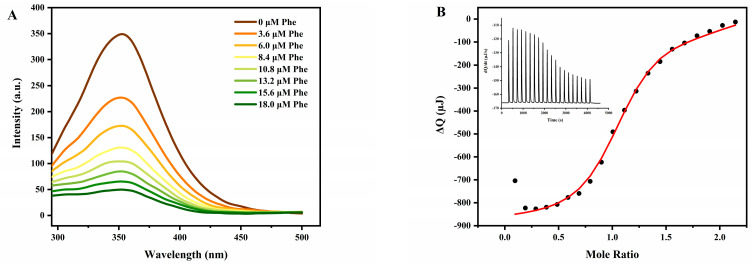
(**A**) The fluorescence spectra of pancreatic lipase with increasing concentration of pheophytin (Phe) and (**B**) ITC plot derived from integrated heat with pheophytin concentration. The solid red line was the corrected result for the heat with a fitting model, while binding site *n* ≈ 1, *K_a_* = (4.38 ± 0.08) × 10^7^ M^−1^ and Δ*H* = −1137.01 ± 0.76 (kJ/mol). Inset: raw data obtained for continuous injection of pheophytin to pancreatic lipase. Data were shown as mean ± SD of three replicates.

**Figure 6 nutrients-14-01749-f006:**
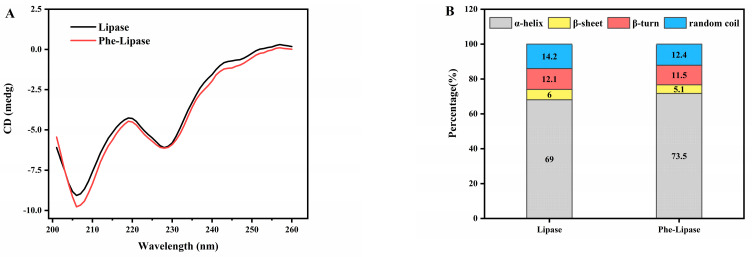
(**A**) Circular dichroism spectra (CD) and (**B**) secondary structure fractions of pancreatic lipase within the absence/presence of pheophytin (Phe). Data were shown as mean ± SD of three replicates.

**Figure 7 nutrients-14-01749-f007:**
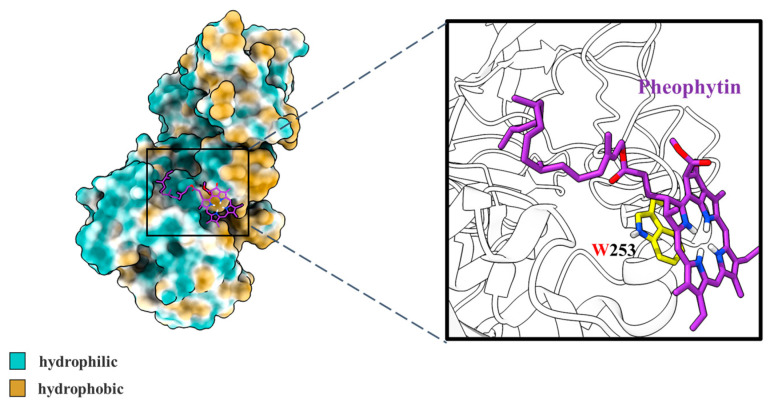
Conformational analysis of molecular docking results for pheophytin and pancreatic lipase. The structure of pheophytin was drawn as sticks, while the structure of lipase was drawn as the surface. The hydrophilic and hydrophobic groups of pancreatic lipase were painted cyan and yellow, respectively. The solid frame area presented a close-up view of the binding site for pancreatic lipase and pheophytin.

**Table 1 nutrients-14-01749-t001:** Kinetics parameters of FFA release under stimulated adult and infant intestinal digestion in vitro.

Group	Apparent Rate Constant *k* (×10^−2^ s^−1^)	Regression Coefficient (R^2^)
AL	3.76	0.9985
AL-CL	3.06	0.9569
AL-CH	3.67	0.9699
EL	4.11	0.9436
EL-CL	2.22	0.9905
EL-CH	3.76	0.9985

**Table 2 nutrients-14-01749-t002:** Fatty acid composition (%) after in vitro simulated gastrointestinal digestion.

Fatty Acids	AL	AL-CL	AL-CH	EL	EL-CL	EL-CH
C12:0	0.04 ± 0.01 a	0.03 ± 0.01 a	0.03 ± 0.00 a	0.03 ± 0.01 a	0.03 ± 0.01 a	0.02 ± 0.00 a
C14:0	0.14 ± 0.00 b	0.14 ± 0.00 b	0.15 ± 0.01 a	0.10 ± 0.00 c	0.11 ± 0.00 c	0.11 ± 0.00 c
C15:0	0.05 ± 0.00 c	0.05 ± 0.00 b	0.06 ± 0.00 a	0.02 ± 0.00 e	0.03 ± 0.00 d	0.04 ± 0.00 c
C16:0	15.01 ± 0.53 b	15.85 ± 0.18 ab	16.4 ± 0.77 a	12.31 ± 0.05 c	12.44 ± 0.11 c	12.39 ± 0.03 c
C16:1	0.17 ± 0.00 b	0.17 ± 0.00 ab	0.18 ± 0.00 a	0.12 ± 0.00 d	0.13 ± 0.00 c	0.12 ± 0.00 cd
C17:0	0.23 ± 0.01 b	0.24 ± 0.00 ab	0.25 ± 0.01 a	0.15 ± 0.01 c	0.15 ± 0.00 c	0.16 ± 0.00 c
C18:0	8.12 ± 0.31 b	8.50 ± 0.13 ab	8.88 ± 0.48 a	5.71 ± 0.04 c	5.67 ± 0.08 c	5.71 ± 0.02 c
C18:1n9c	21.85 ± 0.16 a	21.47 ± 0.06 ab	21.24 ± 0.29 b	21.49 ± 0.38 ab	21.67 ± 0.08 ab	21.46 ± 0.01 ab
C18:2n6c	46.51 ± 0.57 c	45.01 ± 0.31 d	43.37 ± 0.95 e	51.66 ± 0.30 a	51.03 ± 0.17 ab	49.99 ± 0.04 b
C18:3n3	6.08 ± 0.23 d	6.56 ± 0.03 c	7.52 ± 0.18 b	6.85 ± 0.07 c	7.61 ± 0.03 b	8.83 ± 0.04 a
C20:0	0.47 ± 0.04 ab	0.50 ± 0.03 a	0.48 ± 0.04 ab	0.41 ± 0.03 c	0.42 ± 0.02 bc	0.42 ± 0.03 bc
C20:1	0.17 ± 0.01 b	0.19 ± 0.01 b	0.24 ± 0.00 a	0.23 ± 0.01 a	0.23 ± 0.00 a	0.23 ± 0.01 a
C21:0	0.11 ± 0.01 ab	0.13 ± 0.02 a	0.09 ± 0.01 bc	0.09 ± 0.01 bc	0.09 ± 0.00 bc	0.08 ± 0.01 c
C20:4n6	0.12 ± 0.01 a	0.11 ± 0 a	0.12 ± 0.02 a	0.04 ± 0.01 b	0.06 ± 0.01 b	0.05 ± 0.00 b
C22:0	0.59 ± 0.02 a	0.61 ± 0.05 a	0.61 ± 0.07 a	0.45 ± 0.02 b	0.06 ± 0.00 c	0.06 ± 0.01 c
C22:1n9	0.15 ± 0.01 b	0.23 ± 0.02 a	0.13 ± 0.02 b	0.15 ± 0.06 b	0.10 ± 0.00 b	0.11 ± 0.01 b
C24:0	0.25 ± 0.01 a	0.22 ± 0.00 ab	0.20 ± 0.00 b	0.21 ± 0.03 ab	0.19 ± 0.02 b	0.18 ± 0.03 b

Values were shown as the means ± SD of three replicates. The same letter in the same line indicated no significant differences (*p* > 0.05) and different letters showed significant difference (*p* < 0.05).

## Data Availability

Data are available from the corresponding author on reasonable request.

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
