# Peer review of "Chlorophyll Inhibits the Digestion of Soybean Oil in Simulated Human Gastrointestinal System"

_nutrients, 2022, doi:10.3390/nu14091749_

Round 1
Reviewer 1 Report
The authors have investigated the effects of chlorophyll on the digestive characteristics of lipid under an in vitro simulated adult and infant gastrointestinal system. However, numerous problematic issues need to be clarified.
Specific Comments:
- There are some conceptual mistakes along the paper. Dietary fats can differ in their energetic contributions and metabolic effects, making necessary the reevaluation of dietary recommendations. In this sense, ketogenic diets are an example of how a change in the amount and/or profile of fat consumption could result in weight loss. Although it is not the main idea of the paper, they should be corrected.
- Some references are not adequately used, such as references 2 and 5.
- Authors cited along the text should follow the same style; in some cases, many authors are shown (for example, Li, Lu, Wang, Hu, Liao and Zhang [31]), and, in other cases, it is shown “et. al” (for example, Minekus, et al. [32]).
- Introduction section should not include conclusions.
- The aims are not clearly presented. The hypothesis that chlorophyll could prevent obesity by reducing the digestion of lipids has not been tested.
- Authors indicate that emulsions have been prepared w/w. However, some of them have been prepared with PBS. In addition, how emulsions with different concentrations of chlorophyll have been prepared should be clearly presented.
- In Statistical Analysis:
- An important reservation is that chlorophyll does not show a dose-response effect, suggesting that doses have not been adequately selected. Please, explain.
- Figure 1: Data should be shown as mean± SD, indicating statistically significant differences.
- Data showed in Table 1 are also presented in the text. Please, remove.
- In tables and figures, the group to which the statistical significance refers each letter must be indicated.
- The main reservation is that many conclusions are not supported by the results. Without being an exhaustive list, from the evidence of a decrease in the mean particle sizes in adult intestinal vs gastric digestion (not in infants) cannot be concluded that it was due to the hydrolysis of triglycerides and their emulsification to form micelles. Even those this effect was not observed in infants. Similarly, from the evidence that chlorophyll can inhibit lipase activity, it cannot be concluded that this effect is due to the presence of pheophytin. Therefore, the paper results very speculative.
- A native should revise the English style. In many cases, the grammatically errors have changed the meaning of the sentences.
- The authors should consider an overall re-evaluation for the manuscript.
Reviewer 2 Report
The reviewed paper presented the effects of chlorophylls on the lipid digestion and metabolism under in vitro conditions, leading also to the identification of a promising anti-lipase derivative of chlorophyll, namely pheophytin. The study is comprehensive, with a careful design and could be of interest for the readers of Nutritients.
Recommendations:
L17: What do authors mean with “high energy density”?
L23: Which emulsion?
L27: Define the parameter for which you give the value. Which constant you mean
L35: lipid or lipids? In my opinion should be in plural. Please check it and correct all over the manuscript.
L53: suppressing physical food intake -> do authors mean that reduce/modulate the appetite?
L60: Needs rephrasing
L62: horneri should be in italics
L81: Is it indeed a ‘discovery’ of the natural inhibitor?
L115: Format the references style at this line
L116: The authors should detail what SGF contains.
L172: Specify the full name for C11.
L249, L252 – you do not have the explain again the codes for your samples.
L271: Check formatting for ‘2’in equation 2.
L398: The latin name of the plant should be in italics
Round 2
Reviewer 1 Report
The authors have reviewed the manuscript, which has improved significantly. They have addressed many of questions. However, some problems persist.
Please see attachment.
